# All-optical control of spin in a 2D van der Waals magnet

**Maciej Dąbrowski** [1] ✉, **Shi Guo**[1], **Mara Strungaru**[2], **Paul S. Keatley** [1],
**Freddie Withers** [1], **Elton J. G. Santos** [2,3,4] ✉ **& Robert J. Hicken** [1] ✉

Two-dimensional (2D) van der Waals magnets provide new opportunities for control of magnetism at the nanometre scale via mechanisms such as strain, voltage and the photovoltaic effect. Ultrafast laser pulses promise the fastest and most energy efficient means of manipulating electron spin and can be utilized for information storage. However, little is known about how laser pulses influence the spins in 2D magnets. Here we demonstrate laser-induced magnetic domain formation and all-optical switching in the recently discovered 2D van der Waals ferromagnet $CrI_3$. While the magnetism of bare $CrI_3$ layers can be manipulated with single laser pulses through thermal demagnetization processes, all-optical switching is achieved in nanostructures that combine ultrathin $CrI_3$ with a monolayer of $WSe_2$. The out-of-plane magnetization is switched with multiple femtosecond pulses of either circular or linear polarization, while single pulses result in less reproducible and partial switching. Our results imply that spin-dependent interfacial charge transfer between the $WSe_2$ and $CrI_3$ is the underpinning mechanism for the switching, paving the way towards ultrafast optical control of 2D van der Waals magnets for future photomagnetic recording and device technology.

Manipulation of magnetic order using ultrashort laser pulses, via effects such as ultrafast demagnetization[1–3] and all-optical switching (AOS)[4–8], has immense potential for information processing on ultrafast timescales. AOS has been realized principally via thermal processes, such as exchange of angular momentum driven by ultrafast optical heating[4,9–12], and preferential domain nucleation and growth driven by magnetic circular dichroism (MCD)[5,6,13,14], resulting in helicity-independent single-pulse toggle switching, and multi-pulse helicity-dependent switching, respectively. More recently, switching processes utilizing non-thermal changes to magnetic anisotropy[15], hot electrons[16], and interlayer exchange coupling[17] have been explored. Another promising means of manipulating spins with laser pulses is based upon spin-selective charge transfer[18]. This approach is one of the fastest and most coherent, allows for transient changes between antiferromagnetic and ferromagnetic

order[18,19], and can modify the magnitude of the local moments[20]. However, it cannot lead to AOS[18]. Furthermore, in conventional metallic magnets, the transfer of excited electrons can only persist on the timescale of the optical laser pulse length, owing to the screening of the Coulomb interaction. Two-dimensional (2D) van der Waals (vdW) materials such as the transition metal dichalcogenides (TMDCs) offer unique opportunities for spin-dependent charge transfer associated with excitons in $K$ valleys, with the distribution of excitons between the valleys being controlled by the polarization of the light with which the excitons are generated. Excitonic spin-valley polarization in TMDC monolayers can persist for several tens of picoseconds[21,22] up to tens of nanoseconds[23], while efficient spin-dependent charge transfer between stacked 2D flakes can be realized experimentally[24]. Recently, particular attention has been devoted to heterostructures assembled from TMDCs and 2D

[1]Department of Physics and Astronomy, University of Exeter, EX4 4QL Exeter, UK. [2]Institute for Condensed Matter Physics and Complex Systems, School of Physics and Astronomy, The University of Edinburgh, EH9 3FD Edinburgh, UK. [3]Higgs Centre for Theoretical Physics, The University of Edinburgh, EH9 3FD Edinburgh, UK. [4]Donostia International Physics Center (DIPC), 20018 Donostia-San Sebastián, Spain. ✉e-mail: m.k.dabrowski@exeter.ac.uk; esantos@ed.ac.uk; r.j.hicken@exeter.ac.uk

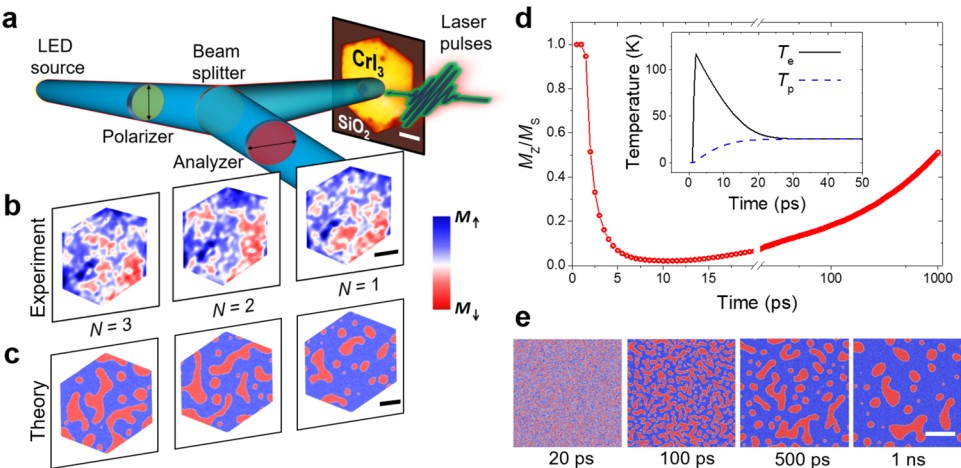

**Fig. 1 | Demagnetization and optically induced domain formation in CrI₃ (sample 1). a** Schematic of the experimental setup and optical image of a ~40 nm bulk CrI₃ flake on Si/SiO₂ (sample 1). Magnetic domains are imaged in a wide-field Kerr microscope (WFKM), by detecting polarization changes of the reflected light due to the polar Kerr effect, following excitation with laser pulses. **b** Formation of domain structure after consecutive single laser pulses ($N = 1, 2, 3$) with 1.88 eV photon energy, linear polarization, ~30 fs pulse duration and fluence $F = 3.4$ mJ/cm². The blue (red) color scale represents the Kerr signal corresponding to the magnetization orientation $M_\uparrow(M_\downarrow)$. The measurements were performed at 20 K. The scale bar has 5 μm in length. **c** Simulated domain structures after the application of consecutive laser pulses using atomistic simulation methods. **d** Time evolution of the spatially averaged out-of-plane magnetization following application of the $N = 1$ laser pulse; (Inset) Electron $T_e$ and phonon $T_p$ temperatures calculated from the two-temperature model. **e** Evolution of magnetic domains following the application of an ultrafast laser pulse at different time steps. The scale bars in **c** and **e** have 100 nm lengths.

magnets, where spin-dependent charge transfer has revealed unprecedented novel functionalities[25–29].

One of the most interesting and widely studied 2D magnets[30,31] is the semiconducting ferromagnet CrI₃, in which the magnetic moments are oriented out-of-plane with either ferromagnetic or antiferromagnetic order, depending upon the number of layers[31]. CrI₃ has shown a number of novel functionalities in terms of intrinsic giant magnetoresistance[32,33], magnetic order tunable via strain engineering[34] and external bias[35–37], hybrid magnetic domain walls[38], exchange-driven magnetostriction[39], and propagation of 2D magnons[40]. Moreover, the incorporation of CrI₃ into heterostructures with TMDCs has opened a new vista for semiconductor electronic and spintronic applications. Due to strong proximity effects, valley polarization and valley Zeeman splitting within a WSe₂ layer can be controlled by the direction of the magnetization of an adjacent CrI₃ layer[25,41,42]. Type-II band-alignment between WSe₂ and CrI₃ allows for spin-dependent charge transfer[27,43] providing additional control of spin and valley polarization[25,26,41,43]. Despite the extraordinary functionalities of CrI₃/WSe₂, attention has primarily concentrated on manipulating the properties of the TMDC via the adjacent 2D magnet. Indeed, the exploration of the opposite scenario, that is, how the presence of the TMDC can be utilized to optically control the magnetic properties of the 2D magnet, is largely unknown.

Here, we use a monolayer of the WSe₂ to manipulate the magnetic properties of the CrI₃ via spin-dependent charge transfer and demonstrate AOS within a magnetic vdW heterostructure. To probe the magnetic domains of CrI₃, we employed wide-field Kerr microscopy (WFKM) in a polar geometry sensitive to out-of-plane magnetization. The sample illumination was linearly polarized, while polarization changes of the reflected light due to the polar Kerr effect were detected as intensity changes using a nearly crossed analyzer and quarter-waveplate (Fig. 1a, see the "Methods" section). The samples were excited by ~30 fs laser pulses and subsequently imaged by WFKM after exposure to one or more pulses. In all measurements a pump spot diameter of 90 μm ($1/e^2$ intensity) was used, which was much larger than the lateral dimensions of the flakes studied, to ensure homogeneous optical excitation across the sample. Remanent magnetic states were prepared by first applying a saturating out-of-plane

magnetic field generated by an air-core electromagnetic coil, with optical pumping then being performed at zero fields.

## Results

Measurements were first made on an isolated ~40 nm-thick bulk flake of CrI₃ (sample 1) that possessed primarily ferromagnetic order, as indicated by hysteresis loop measurements (see Supplementary Fig. S6). Starting from an initial uniform mono-domain remanent state with the magnetization **M** pointing up (↑) from the sample surface, domain formation is observed with disordered $M_\uparrow$ and $M_\downarrow$ sub-domains that are randomly rearranged after each consecutive ($N$th) laser pulse (Fig. 1b). Similar domain patterns are observed for exposure to multiple pulses and different pump polarizations (Supplementary Fig. S7), with such behavior being ascribed to domain formation following laser-induced thermal demagnetization[6]. Note that all images presented in this work are acquired for the final magnetization state which remains the same until the sample is exposed either to more optical pulses or to an external magnetic field.

To gain more insight into the optically induced demagnetization process, multiscale atomistic spin dynamics simulations[38,44] were performed of CrI₃ excited by an ultrafast laser pulse (see Section S6 for details). The effect of the laser pulse has been modeled via coupled equations that describe the evolution of the electron and phonon temperatures (known as the two-temperature model), while the dynamics of the spin system are treated by means of a Langevin approach. Starting from a uniform mono-domain state, laser pulses nucleate magnetic domains, which appear rearranged after each consecutive ($N$th) pulse, as observed in the experiment (Fig. 1c). The simulations show smaller domains since the spin configurations are extracted 1 ns after application of the laser pulse, while the continued merger of domains occurs on longer timescales. The relatively slow dynamics are in agreement with the evolution of the spatially averaged out-of-plane magnetization (Fig. 1d). Although the electron and phonon temperatures quickly reach equilibrium (inset in Fig. 1d), the magnetization continues to evolve on the nanosecond timescale due to the continuously changing magnetic domain structure (Fig. 1e). The metastability of the magnetic domains in the CrI₃[38] also contributes to the dynamics of the magnetic structure after the laser excitation.

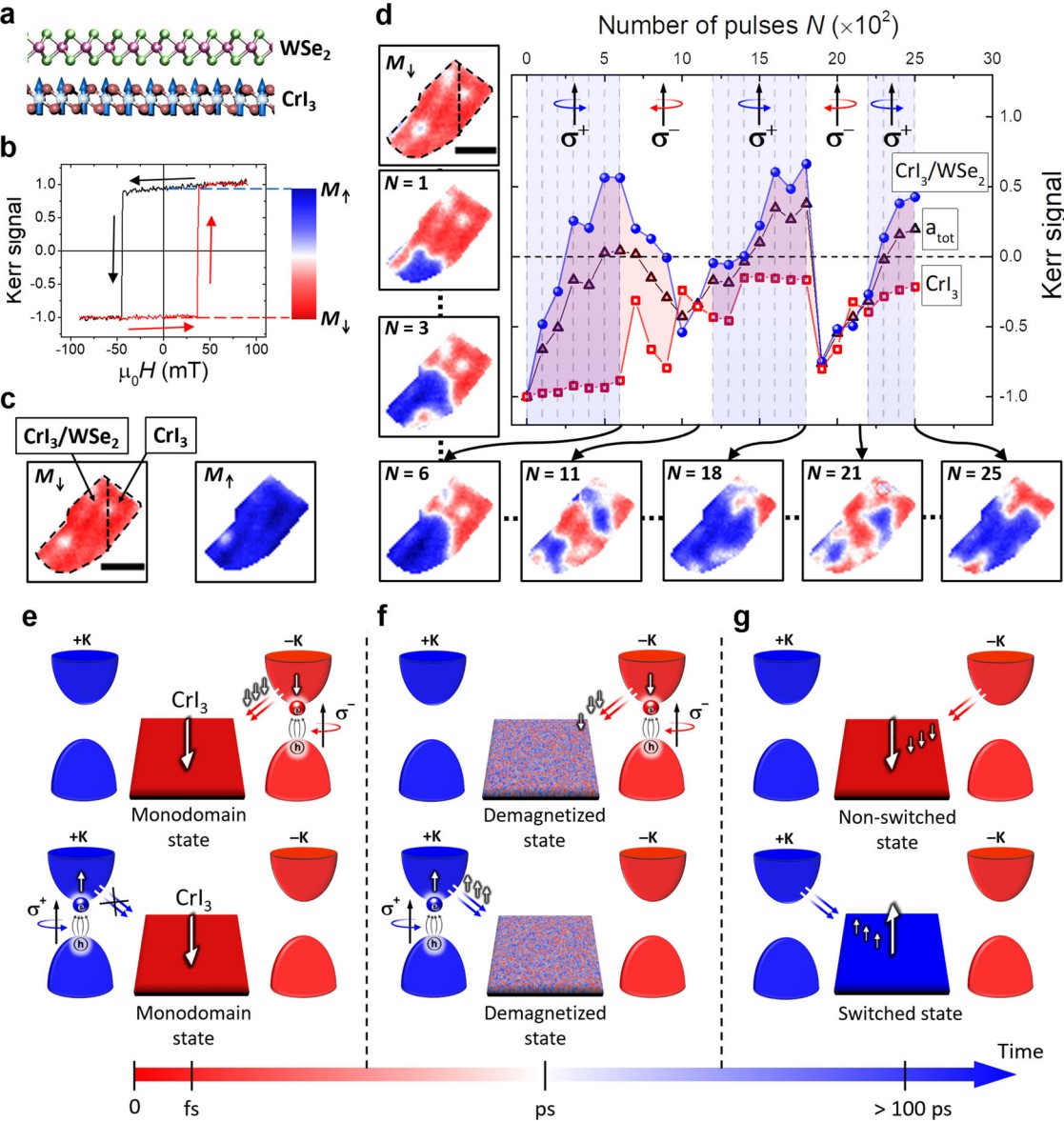

**Fig. 2 | Helicity-dependent all-optical switching (HD-AOS) in a CrI₃(10 nm)/WSe₂(1L) heterostructure (sample 2). a** Schematic of the CrI₃/WSe₂ heterostructure. **b** Polar MOKE hysteresis loop with the Kerr signal integrated over the entire area of the flake ($a_{tot}$). The blue-red color scale represents a remanent Kerr signal ranging between magnetization orientations $M_\uparrow$ and $M_\downarrow$ obtained after saturation at +100 and −100 mT, respectively. **c** Remanent monodomain states $M_\uparrow$ and $M_\downarrow$. The CrI₃ flake is partially overlapped with a WSe₂ monolayer, as indicated by the black dashed lines in the $M_\downarrow$ domain state image. **d** Changes to the Kerr signal after optical pumping with circular polarization $\sigma^+$ (shaded areas with dashed lines) and $\sigma^-$ (transparent areas). $N$ bunches of $10^2$ pulses are applied. The Kerr signal is extracted from the domain structure images and is plotted for different parts of the

flake: total area $a_{tot}$, CrI₃/WSe₂ only, and CrI₃ only. The measurements were performed at 35 K, with pump photon energy 1.67 eV, -30 fs pulse duration and fluence $F = 6.9$ mJ/cm². The scale bar has 5 µm in length. **e–g** Schematic of the HD-AOS mechanism in the CrI₃/WSe₂ heterostructure for circular light polarizations $\sigma^-$ (top panel) and $\sigma^+$ (bottom panel). **e** Valley-dependent optical selection rules for monolayer WSe₂ and spin-dependent charge transfer between the WSe₂ and CrI₃ for the magnetic moment orientation of the CrI₃ pointing down. The optical helicity is defined by the wavevector pointing up, i.e., out of the surface plane and opposite to the CrI₃ magnetization direction. **f** Schematic of the spin-dependent charge transfer during demagnetization. **g** Outcome of the charge transfer on the AOS.

It is known that a ferromagnetic van der Waals heterostructure formed by monolayer (1L) WSe₂ and thin CrI₃ (Fig. 2a) exhibits spin-dependent charge transfer[25–27,41], which provides an additional mechanism for manipulating the magnetic order[43]. According to the valley-dependent optical selection rules for monolayer WSe₂, for photon energies around or above the WSe₂ band edges, circularly polarized light $\sigma^+$ and $\sigma^-$ should populate the $K$ valleys of the conduction band as $|+K, \uparrow\rangle$ and $|-K, \downarrow\rangle$, respectively, where $\uparrow$ ($\downarrow$) represents the magnetic moment orientation (shown by white arrows in Fig. 2e). The charge transfer in CrI₃/WSe₂ is helicity-dependent and in general is allowed only when the photo-excited electron spin in the

WSe₂ has the same orientation as the spin in the CrI₃ conduction band.

To explore the possibility of AOS induced by charge transfer, bunches of circularly polarized pulses were applied to a structure (sample 2) in which a 10 nm-thick CrI₃ flake was partially overlapped with a WSe₂ monolayer, as indicated by the black dashed lines in the remanent $M_\downarrow$ domain state image in Fig. 2c. The hysteresis loop in Fig. 2b, obtained by integrating the Kerr signal over the whole CrI₃ flake, confirms the ferromagnetic order and magnetization reversal via a single switching event. Furthermore, the magnetization reversal is not affected by the WSe₂. Since the presence of the WSe₂ does not

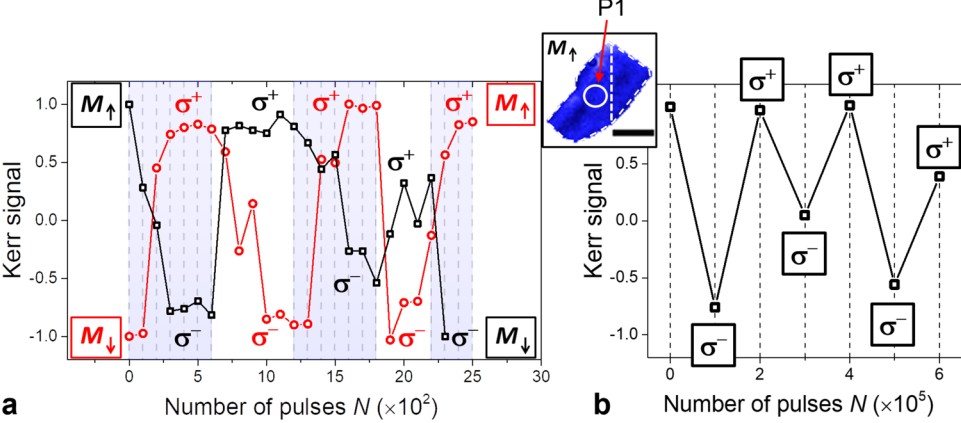

**Fig. 3 | HD-AOS within circular region P1 in the center of the heterostructure CrI$_3$(10 nm)/WSe$_2$(1L) (sample 2). a** HD-AOS induced by $N$ bunches of $10^2$ pulses with circular polarization alternating between $\sigma^+$ and $\sigma^-$. The plot shows the effect of the optical pumping for two different initial remanent states, $M_\uparrow$ (black squares) and $M_\downarrow$ (red circles). **b** HD-AOS for $N$ bunches of $10^5$ pulses alternating between $\sigma^-$ and $\sigma^+$ circular polarization. The Kerr signal in **a** and **b** is extracted from the 2 μm wide circular spot P1 outlined with a white circle in the $M_\uparrow$ domain structure image shown in the inset. The measurements were performed at 35 K, with pump photon energy 1.67 eV, -30 fs pulse duration and fluence $F = 6.9$ mJ/cm$^2$. The scale bar has 5 μm in length.

affect either the coercive field or the remanent mono-domain state of the CrI$_3$, one can conclude that the magnetic ground state is not affected by the WSe$_2$, and therefore that differences in the evolution of the domain structure observed with and without WSe$_2$ can be attributed solely to optical pumping. Starting from the uniform remanent ground state $M_\downarrow$, the sample was exposed to a sequence of laser pulses with 1.67 eV photon energy, centered around the WSe$_2$ main photoluminescence peak, which is assumed to result from a positively charged trion state in the WSe$_2$ due to the type-II band alignment[41]. Figure 2d displays the Kerr signal integrated over the entire area of the flake $a_{\text{tot}}$ (black triangles), the part with CrI$_3$/WSe$_2$ (blue circles), and the isolated CrI$_3$ (red squares). The optical pulses had circular polarization, $\sigma^+$ or $\sigma^-$, and were applied in bunches of $10^2$. In the case of the CrI$_3$/WSe$_2$ heterostructure, optical pumping with $\sigma^+$ polarization progressively reorients most of the domains from the initial $M_\downarrow$ remanent state into the $M_\uparrow$ state. After a total of $6 \times 10^2$ pulses, the population of $M_\downarrow$ domains can be partially restored by changing the helicity to $\sigma^-$. Therefore, by alternating the helicity, partial switching backwards and forwards between the $M_\downarrow$ and $M_\uparrow$ states can be achieved. In contrast, the part of the sample without WSe$_2$ retains the majority of its domains in the same magnetization orientation (the Kerr signal does not cross zero) regardless of the optical polarization. The fact that substantial changes to the magnetization direction are only observed in the part of the CrI$_3$ overlapped with the WSe$_2$ suggests that spin-polarized charge transfer may play a role. To explain the helicity-dependent reorientation of the magnetization, let us consider the $M_\downarrow$ initial state where, according to the valley-dependent optical selection rules, the charge transfer should be allowed for $\sigma^-$, but not for $\sigma^+$ (Fig. 2e). Excitation by an ultrashort laser pulse leads to instantaneous creation of hot electrons, followed by equilibration of the electronic system at elevated temperatures. During this process of demagnetization, a new channel for the charge transfer is opened, since $M_\downarrow$ and $M_\uparrow$ magnetization states become equally likely (see Fig. 2f, where the demagnetized state is represented by a simulated domain structure at 20 ps after laser excitation, as in Fig. 1e), and electrons from both $K$ valleys can be transferred to the CrI$_3$. Thus, when pumping the $M_\downarrow$ state with $\sigma^+$ (and correspondingly $M_\uparrow$ with $\sigma^-$), the charge transfer will occur primarily for the electron spin orientation antiparallel to the initial spin state in the CrI$_3$, which induces reorientation of the CrI$_3$ magnetization. Once the magnetization switches, the charge transfer is expected to affect the magnetization only after the optical helicity is again reversed, i.e., only after the spin orientation of the electrons

excited in the WSe$_2$ is again antiparallel to the initial state in the CrI$_3$. This is exactly what is observed in the experiment.

The magnetization within the CrI$_3$/WSe$_2$ flake does not switch uniformly, and only partial AOS is observed in Fig. 2d, due to a general tendency for domain formation in CrI$_3$[38,41]. A plausible interpretation is that, after each optical pulse, as the sample cools and remagnetization takes place, domains are formed in order to minimize the magneto-static energy, so that the mono-domain state cannot be preserved. The tendency for the system to form domains is also confirmed by the simulations (Fig. 1 and Section S6). This also explains why the efficiency of the AOS decreases as the number of pulses and switching events increases. Furthermore, although the CrI$_3$ flakes are assumed to be spatially uniform, localized defects and irregularities in the magnetization and stray field are always present[45], which will again favor domain creation during the thermal cycling induced by optical pumping. Finally, it should be stressed that, until now, AOS has been explored primarily in thin continuous films, where the switched regions exposed to optical pulses were not obstructed by any μm-sized boundaries. Even then, in most cases, full AOS induced by multiple pulses could only be achieved by sweeping the beam across the sample[5–7], and only for domains with a size larger than that of the laser spot[46].

To minimize the impact of inhomogeneities at the physical boundaries of the flake, let us consider the region P1, a 2 μm diameter circle outlined by a white solid line in Fig. 3, within the interior of the region covered by WSe$_2$. Figure 3a shows exactly the same experiment as Fig. 2, but for the Kerr, signal integrated over P1. The initial remanent states $M_\downarrow$ (red circles) and $M_\uparrow$ (black squares), were optically pumped with opposite helicities $\sigma^+$ and $\sigma^-$, respectively. By alternating the optical helicity, full switching between the two oppositely oriented out-of-plane magnetization orientations $M_\downarrow$ and $M_\uparrow$ can be achieved. Notably, pumping with the $\sigma^+$ ($\sigma^-$) polarization always results in the $M_\uparrow$ ($M_\downarrow$) final state. Furthermore, the switching requires a number of bunches of $10^2$ pulses before the effect saturates, and further pumping does not change the final state of the magnetization. These results demonstrate that full and reproducible helicity-dependent AOS (HD-AOS) is observed within the P1 region, i.e., in the center of the heterostructure, further away from the edges of the flake and their associated defects.

The number of pulses per bunch was then increased to $10^5$ to determine whether AOS can be achieved with just a single bunch. Figure 3b shows the Kerr signal extracted from the area P1, following pumping with single bunches where the circular polarization

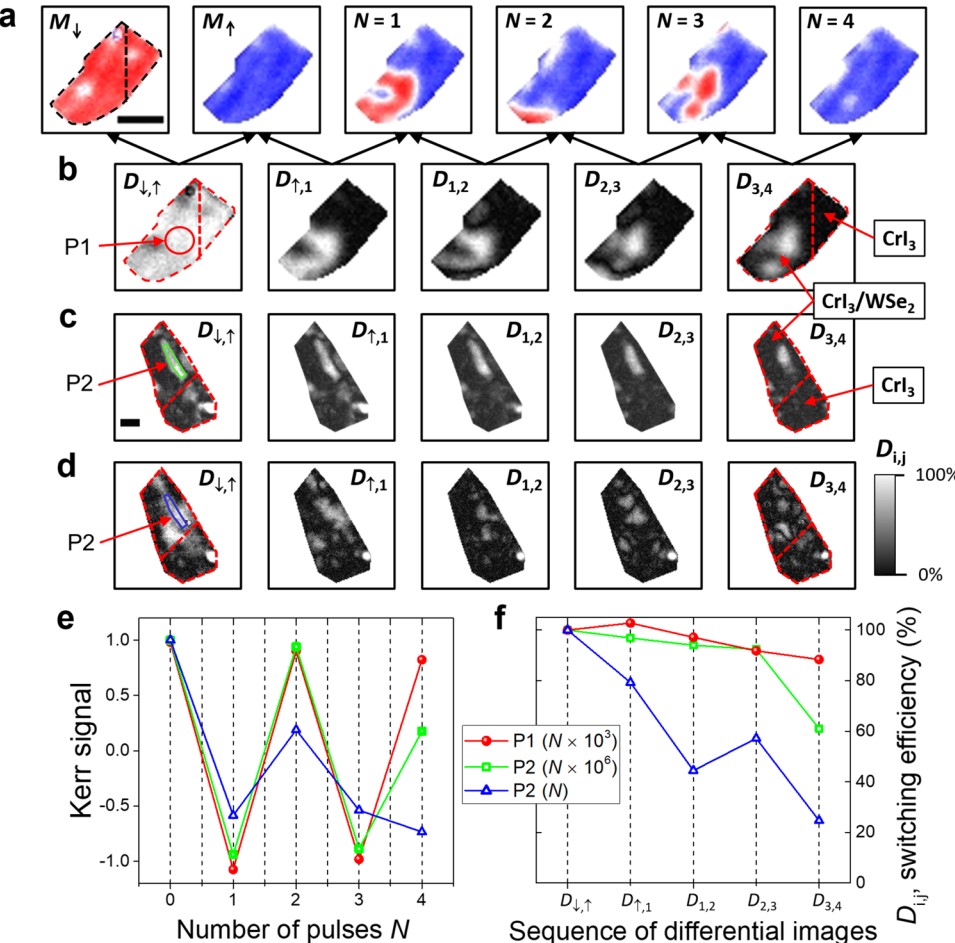

**Fig. 4 | Demonstration of toggle AOS with linearly polarized pulses for a CrI$_3$(10 nm)/WSe$_2$(1L) heterostructures (samples 2 and 3). a** The remanent $M_\downarrow$ and $M_\uparrow$ mono-domain states for sample 2, and the $M_\uparrow$ remanent state after exposure to $N$ bunches of $10^3$ pulses. **b** Differential Kerr signal $D_{i,j} = |(M_i - M_j)|/(|M_i| + |M_j|)$ images extracted from the images in **a**. Differential images for sample 3 pumped with (**c**) $N$ bunches of $10^6$ pulses and **d** single pulses. **e, f** Kerr signal and differential Kerr signal $D_{i,j}$ (switching efficiency) integrated over the areas P1 (sample 2) and P2 (sample 3). The measurements were performed at 35 K, with pump photon energy 1.67 eV, -30 fs pulse duration and fluences (**a, b**) $F = 6.9$ mJ/cm$^2$, (**c**) $F = 6.4$ mJ/cm$^2$, and **d** $F = 7.1$ mJ/cm$^2$. The scale bars have 5 μm in length.

alternates between $\sigma^+$ and $\sigma^-$. Switching between the $M_\uparrow$ and $M_\downarrow$ states can be achieved in a similar manner as in Fig. 3a, although with slightly worse reproducibility. Qualitatively, however, the switching process still follows the same helicity dependence. In general, because the time between each single pulse (1 μs in this case) might not be sufficient for the sample to return to its equilibrium state, it is expected that pumping with a different number of pulses per bunch will result in a different AOS efficiency. Further optimization of the laser parameters, such as the repetition rate and pulse duration, should allow for better control of the AOS process in structures based on CrI$_3$ flakes.

We next explore whether switching of the CrI$_3$ spins is limited to excitation by circularly polarized bunches. Figure 4a shows the domain structure for the same CrI$_3$/WSe$_2$ heterostructure as shown in Figs. 2 and 3 (sample 2), but following exposure to linearly polarized pulses. Similar to the case of optical pumping with circularly polarized light, substantial changes to the Kerr signal occur only in the part of the sample where the WSe$_2$ overlaps the CrI$_3$. To follow the changes of magnetization orientation within domains, it is useful to consider the differential Kerr signal $D_{i,j} = |(M_i - M_j)|/(|M_i| + |M_j|)$, i.e., the difference between two consecutive images, as defined in Fig. 4b. The differential signal is normalized to the differential remanence, $D_{\downarrow,\uparrow}$, that corresponds to 100% reversal, i.e., full switching between mono-domain states $M_\downarrow$ and $M_\uparrow$. The differential signal can therefore be treated as a measure of the switching efficiency achieved by successive bunches of pulses. From the differential images, it is immediately obvious which

regions of the flake switch and which are unchanged by optical pumping. As shown in Fig. 4b, only the CrI$_3$/WSe$_2$ exhibits switching, and similar to the case of circular pump polarization, the switching is non-uniform and domains are created. Nevertheless, the P1 region of sample 2 switches consistently, as quantified in Fig. 4e (red circles) by the spatially integrated Kerr signal and switching efficiency $D_{i,j}$. Hence, we demonstrate helicity-independent toggle switching in the CrI$_3$/WSe$_2$ heterostructure.

In Fig. 4c and d measurements are presented for a further heterostructure (sample 3) in which 10 nm-thick CrI$_3$ is partially overlapped with monolayer WSe$_2$, i.e., for a flake with the same thicknesses as sample 2 but with larger lateral dimensions. In addition, sample 3 exhibits a spatially inhomogeneous differential remanent signal $D_{\downarrow,\uparrow}$, indicating that the maximum available field of 100 mT is only able to reverse the magnetization in selected regions of the flake (see Supplementary Fig. S14). This inhomogeneity is also reflected in the optically induced domain structure formation. Figure 4c shows the differential images obtained by pumping with bunches of $10^6$ linearly polarized pulses. While the effect of the first bunch ($D_{\uparrow,1}$) is to switch nearly the same area as the magnetic field ($D_{\downarrow,\uparrow}$), the switching efficiency decreases with successive bunches with a reduction of the switched area. Note that, similar to the case of sample 2, the switching can only be observed in the part of the CrI$_3$ flake overlapped with the WSe$_2$. For more quantitative analysis, the Kerr signal and $D_{i,j}$ integrated over the area P2, mimicking the shape of the largest domain in the

remant state $D_{\downarrow,\uparrow}$ in Fig. 4c, has been plotted (green squares) in Fig. 4e. This demonstrates that sample 3 also exhibits AOS in response to linearly polarized pulses and that the switching efficiency is comparable to sample 2. Finally, the effect of pumping with single laser pulses is shown in Fig. 4d. Single pulses are less efficient in inducing AOS (see blue triangles in Fig. 4e) and smaller domains are created after each consecutive pulse. AOS of similar efficiency using a single laser pulse was also observed in sample 2 (Supplementary Fig. S12). Single pulse AOS requires more fluence ($F > 7$ mJ/cm$^2$), increasing the risk of sample damage (see Supplementary Fig. S17), and so was not extensively explored. AOS occurs under very similar laser parameters for both samples 2 and 3, but a complete set of data was acquired only for sample 3, due to sample degradation by repeated exposure to laser pulses and by thermal cycles during cooling–warming up processes (see laser fluence and repetition rate dependence in Supplementary Fig. S15).

There are several mechanisms that could potentially lead to the AOS in isolated CrI$_3$, such as magnetic circular dichroism (MCD)[6], the inverse Faraday effect (IFE)[47], and the very recently proposed switching through resonant coupling to excitonic transitions[48]. The AOS is however only observed in the CrI$_3$/WSe$_2$ part of the sample, so the valley-dependent charge transfer appears essential for switching to occur. Note that for excitation with a photon energy $E = 1.2$ eV, i.e., well below the WSe$_2$ band gap, no switching could be observed, further confirming that the interfacial charge transfer is essential for AOS to occur. Finally, note that all of the alternative effects mentioned above can only result in HD-AOS, i.e., they cannot be achieved with linearly polarized light. The question arises of how to explain the AOS driven by the charge transfer in the case of linearly polarized excitation? Considering the optical transitions within an isolated monolayer of WSe$_2$, linearly polarized excitation should generate a coherent superposition of excitons in both $K$ valleys[49]. However, for the CrI$_3$/WSe$_2$ heterostructure, electron hopping from WSe$_2$ to CrI$_3$ is only allowed when the spin orientation of the electron in WSe$_2$ is the same as that of the lowest-energy unoccupied conduction bands, as shown schematically in Fig. 2e for the $M_{\downarrow}$ initial state. Hence, immediately after optical excitation with linear polarization, and before demagnetization has occurred in the CrI$_3$, the transfer of electrons from the WSe$_2$ with spin parallel to that within the CrI$_3$ will begin, leaving electrons with the spin opposite to that of the CrI$_3$ in the other $K$ valley. Then, as the CrI$_3$ becomes demagnetized, transfer of electrons with either spin type becomes possible (Fig. 2f). However, there will be an imbalance of the remaining electrons in the $K$ valleys, with more electrons having spin opposite to that of the initial CrI$_3$ spin direction (e.g., as shown in Fig. 2f, where some of the spin-down electrons have already been transferred, and there are more spin-up electrons available for transfer during demagnetization and remagnetization). Therefore, the transfer of the remaining spin-polarized electrons will induce remagnetization of the CrI$_3$ in the reversed state. This mechanism assumes that the initial charge transfer for one of the valleys can occur on sub-picosecond timescales (Fig. 2e), before demagnetization of the CrI$_3$ is complete, which is in agreement with the current understanding of the charge transfer dynamics in 2D van der Waals materials[28,50,51]. Finally, we also cannot completely exclude the possibility that the AOS occurs via the exchange interaction between photo-excited spin-polarized carriers within the WSe$_2$ and electrons within occupied bands of the CrI$_3$. In either case, i.e., spin-dependent charge transfer or spin transfer mediated via the exchange interaction, the presence of the WSe$_2$ appears to be crucial for the AOS of the CrI$_3$ spins. Our results demonstrate that although the charge transfer, and therefore AOS, is intrinsically helicity-dependent, toggle switching with linearly polarized light is feasible owing to the different lifetimes for spin-polarized hot electrons in the two $K$ valleys. This unique property, not accessible in previously studied magnetic materials, offers a range of new functionalities. For instance, by applying a dual-pulse excitation scheme[52],

with the first pulse demagnetizing the CrI$_3$ and the second pulse creating excitons in the $K$ valleys, femtosecond and polarization-dependent control of spins in a 2D magnet should be possible.

Both helicity-dependent and helicity-independent AOS have been demonstrated in ultrathin CrI$_3$. While complete switching is achieved with multiple pulses, single-pulse switching is less spatially uniform and not fully deterministic. The results suggest that the AOS is associated with spin-dependent charge transfer across the CrI$_3$/WSe$_2$ interface. The charge transfer is expected to begin within a few to hundreds of femtoseconds[28], and in principle should allow for control of magnetic properties on unprecedented ultrafast timescales. A complete understanding of the AOS mechanism in CrI$_3$/WSe$_2$ will require more sophisticated calculations and time-resolved measurements. However, the present study clearly demonstrates an unexplored and unique aspect of few vdW layer magnetism, in which laser excitations can be used to deliver fast optical control of magnetization processes.

## Methods

### Sample preparation

For heterostructure samples (samples 2 and 3), ~10 nm-thick CrI$_3$ and ~10 nm h-BN were first exfoliated onto a SiO$_2$(300 nm)/Si substrate. Monolayer WSe$_2$ was exfoliated on polydimethylsiloxane (PDMS) films. To prevent contamination, the exfoliation was performed in a nitrogen-filled glove box. The heterostructure was fabricated by means of an all-dry transfer method commonly used in the literature[53]. Firstly, h-BN flakes, and then CrI$_3$ and monolayer WSe$_2$ were picked up in sequence by a stamp consisting of a thin film of polypropylene carbonate (PPC) on PDMS. Then, the whole h-BN/CrI$_3$/WSe$_2$ stack was released on the h-BN flake which was exfoliated onto the SiO$_2$/Si. There were no signs of degradation in the h-BN-sandwiched samples under ambient conditions, and the time taken to transport and mount the sample in the wide-field Kerr microscope (WFKM) was limited to ~10 min. For uncovered bulk CrI$_3$ flake samples (samples 1 and 1A), the CrI$_3$ flakes were exfoliated directly onto the SiO$_2$/Si, and not covered by h-BN.

### Wide-field Kerr microscopy (WFKM)

The polar Kerr effect was used to sense the out-of-plane magnetization in response to either a magnetic field or optical pulses. The sample illumination was linearly polarized, while polarization changes of the reflected light due to the polar Kerr effect were detected as intensity changes using a nearly crossed analyzer, quarter-waveplate, and high sensitivity CMOS camera. For all-optical switching (AOS) experiments, an optical pump beam of variable polarization, pulse duration, and repetition rate was incident at 45° to the sample plane and focused to a 90 μm diameter spot (intensity at $1/e^2$). For most experiments, an optical parametric amplifier (OPA) with output tunable from 650 to 900 nm and with a fixed pulse duration of 30 fs was employed. Measurements were performed at temperatures ranging from 15 to 45 K.

### Atomistic spin dynamics

We model the system through atomistic spin dynamic simulations[38]. The exchange interactions for CrI$_3$ have been previously parameterized from accurate ab initio calculations[44] and contain up to three next-nearest neighbors. We include the effect of the laser pulse via the two-temperature model (2TM) which couples the electron and phonon baths to the spin dynamics. Additional details are included within Supplementary Section 6 and parameters for the two-temperature model are shown in Supplementary Table 1.

## Data availability

All the data supporting the findings of this study are available within the paper and the Supplementary Information and have been

deposited in Open Research Exeter (ORE) repository at https://doi.org/10.24378/exe.4184.

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

## Acknowledgements

The authors acknowledge the Engineering and Physical Sciences Research Council (EPSRC) Grant EP/V048538/1. The Exeter Time-Resolved Magnetism Facility (EXTREMAG—EPSRC Grant Reference EP/R008809/1 and EP/V054112/1) is acknowledged. E.J.G.S. acknowledges computational resources through CIRRUS Tier-2 HPC Service (ec131 Cirrus Project) at EPCC funded by the University of Edinburgh and EPSRC (EP/P020267/1); ARCHER UK National Supercomputing Service (http://www.archer.ac.uk) via Project d429. E.J.G.S. acknowledges the EPSRC Early Career Fellowship (EP/T021578/1), the Spanish Ministry of Science's grant program "Europa-Excelencia" under grant number EUR2020-112238 and the University of Edinburgh for funding support.

## Author contributions

M.D., F.W., and R.J.H. conceived the study of the bilayer structures while E.J.G.S. conceived the simulation approaches; M.D. performed AOS experiments and analyzed the data; S.G. fabricated and characterized the samples, assisted by F.W.; M.S. and E.J.G.S. carried out the theoretical calculations; M.D. and P.S.K. set up the AOS measurements; M.D. prepared the original manuscript with help from M.S., E.J.G.S., and R.J.H.; all authors discussed the results and contributed to the manuscript; R.J.H. supervised the project.

## Competing interests

The authors declare no competing interests.
