## [Peer Review File · Nature Communications]

Reviewers' Comments:

Reviewer #1:

Remarks to the Author:

In the revised manuscript, Dabrowski and co-authors addressed most of the questions raised by reviewers. Scientifically, the most exciting discovery of this paper is that the spin-dependent interlayer charge transfer can be harvested for the all-optical-switching (AOS) of ferromagnetic domains in the VdWs magnets. Although the demonstrated AOS is partial and the detailed switching process is not fully clear yet, these findings open a new route of the AOS in layered magnets, and will certainly stimulate the efforts of interfacial engineering of 2D magnets for better AOS performance. Hence, I recommend the publication of the current manuscript in Nature Communication.

Reviewer #1 (Remarks to the Author):

In the revised manuscript, Dabrowski and co-authors addressed most of the questions raised by reviewers. Scientifically, the most exciting discovery of this paper is that the spin-dependent interlayer charge transfer can be harvested for the all-optical-switching (AOS) of ferromagnetic domains in the VdWs magnets. Although the demonstrated AOS is partial and the detailed switching process is not fully clear yet, these findings open a new route of the AOS in layered magnets, and will certainly stimulate the efforts of interfacial engineering of 2D magnets for better AOS performance. Hence, I recommend the publication of the current manuscript in Nature Communication.

Reply: We thank the reviewer for his/her recommendation to publish our work without any need for further revisions.